# Multi-Field Interference Simultaneously Imaging on Single Image for Dynamic Surface Measurement

**DOI:** 10.3390/s20123372

**Published:** 2020-06-15

**Authors:** Weiqiang Han, Xiaodong Gao, Zhen Chen, Le Bai, Bo Liu, Rujin Zhao

**Affiliations:** 1Institute of Optics and Electronics of Chinese Academy of Sciences, Chengdu 610209, China; 13980523631@139.com (X.G.); chenzhen@ioe.ac.cn (Z.C.); BL511690069@163.com (L.B.); boliu@ioe.ac.cn (B.L.); zrj0515@163.com (R.Z.); 2Key Laboratory of Science and Technology on Space Optoelectronic Precision Measurement, CAS, Chengdu 610209, China; 3University of Chinese Academy of Sciences, Beijing 100149, China

**Keywords:** vibrating mirrors, interferometer, multiple field interference, pulsed laser, laser repeat frequency, multiple imaging, surface measurements

## Abstract

To obtain the dynamic surface of high-frequency vibrating mirrors (VMs), a novel method involving multi-field interference (MFI) pattern imaging on a single image is proposed in this paper. Using multiple reflections and refractions, the proposed method generates three interference patterns at the same time, which improves the traditional time-series methods where a single interference pattern can be obtained at one time. Experimental results show that a series of MFI patterns can be obtained on a single image, with the laser repetition frequency (LRF) ranging from 200 Hz to 10 Hz, and the frame rate of the camera at 10 Hz. Particularly if the LRF (10 Hz) is equal to the frame rate of image, crosstalk is avoided completely, which is particularly desirable in dynamic surface measurement. In summary, the MFI imaging method provides an effective way for VM dynamic surface measurement.

## 1. Introduction

High-frequency vibrating mirrors (VM) [1] are commonly utilized to suppress background light during small infrared target detection. For example, Wang’s research on real-time background deduction using VMs shows that background radiation that is 208 times stronger than the target can be removed in real time [2]. However, the dynamic surface shape of the VM may directly affect the imaging quality of the optical system when the VM is vibrating. The vibration gives rise to a deformation of the mirror surface, which consequently causes the aberration of the wave front that is reflected by the mirror. This wave-front deformation will ultimately affect the imaging properties of optical systems that use VMs [3]. Therefore, it is important to measure the dynamic surface shape of the VM at optical level accuracy, so as to compensate for defects via digital signal processing in real time. Generally speaking, surface measurements are mainly divided into two types [4,5,6,7,8,9]: contact and non-contact methods. The main advantage of contact methods is that they can achieve micron-level accuracy. Their disadvantages are mainly as follows: (1) it is easy to damage the target’s surface; (2) subtle features of complex target surfaces are difficult to obtain; (3) they are slow; and (4) probes are easy to wear, which degrades the measurement accuracy and shortens their service life. Non-contact methods do not require contact with the target, thereby avoiding surface damage, and are faster than contact methods.

Among various non-contact surface measurement methods, optical methods [10,11,12,13,14,15,16] are widely used in surface measurements. In general, optical non-contact methods can be classified into the following methods: laser ranging, structured light and interference methods. Specifically, laser ranging methods obtain objects’ images through direct or indirect measurement of the travel time using a scanning mechanism. The imaging speed is limited by the scanning mechanism, which makes it unsuitable for dynamic measurements. Structured light methods project a prepared pattern image on the object’s surface, and the surface contour modulates this image. Then, an image of the object is reconstructed from the deformed reflection image. Compared with laser ranging and structured light methods, interference methods [17,18,19,20,21,22,23,24,25,26,27,28,29,30,31] are more suitable for dynamic measurements, due to the advantages of real-time performance and high accuracy.

Some interference measurement methods use scattered light on the object’s surface for vibrating surfaces. Chen et al. used electronic speckle pattern interferometry (ESPI) to measure four vibration modes of a corrugated plate excited by the harmonic wave of a 550 Hz speaker [32]. De Veuster et al. used digital speckle pattern interferometry (DSPI) [33] to measure the amplitude of a diaphragm driven by a speaker at a vibration frequency of 1000 Hz. These types of speckle interference methods are mainly aimed at rough surfaces, and are suitable for interferometric imaging measurements of vibration nodes in vibration modes, rather than surface shape measurement.

There are three ways to implement interference for surface shape measurements: fringe scanning [34]; multi-interferogram [35,36,37,38,39,40,41,42,43,44,45,46]; and single-interferogram [47] methods. Fringe scanning methods shift the mirror in the reference arm so that the light intensity at any point in the interferogram is modulated by the sine function; then, the surface to be measured is derived using the Fourier transform. Multi-interferogram methods acquire interferograms with different reference phase positions by moving a scanning mechanism. In fact, both fringe scanning and multi-interferogram methods require a scanning mechanism to obtain properly timed sequence images. As a result, it is impossible to measure the surface of fast-moving VMs due to the fast multi-frame image acquisition required. Compared with fringe scanning and multi-interferogram methods, single interferogram methods can capture one interferogram on a single image quickly, as they do not require a scanning mechanism. However, their accuracy is lower than that of fringe scanning and multi-interferogram methods (the accuracy of single interferogram methods is λ/10 [48], while the accuracy of fringe scanning reaches λ/100 [34] and multi-interferogram methods usually achieve better than λ/50 [48]). Furthermore, single interferogram methods cannot handle closely spaced interference fringes.

All the above methods are time-series methods, which can only obtain one interferogram at a time. Due to the shortcomings of single interferogram methods, it is common to use three interferograms for interferometric measurements. However, for high-frequency VM dynamic surface shape measurements, multiple images obtained at different time points cannot be used for this purpose. In this paper, a novel spatial sequence method based on single-image multi-field interference (MFI) imaging is proposed, which can effectively deal with the above-mentioned problems in existing methods. Using the proposed method, we can obtain three interference patterns from a single image at the same time, which improves the traditional time-series methods where a single pattern is obtained at one time. As a result, a new type of spatial sequencing is introduced. This allows us to avoid the time delay present in fringe scanning and multi-interferogram methods, which will be helpful in capturing the dynamic surface of VM.

## 2. Multi-Field Interference Imaging Setup

The schematic of the setup for applying multi-field interference is shown in Figure 1. First, light emitted from the laser is shaped through the cylindrical lens (CL), and then passes through a linear polarizer, which transforms partially polarized light to linearly polarized light (p-wave). Second, the linear polarizer, a polarization beam splitter (PBS) and a quarter wave plate (QWP) together act as a spatial light circulator. In this case, the transmitted and returned light can be separated easily, which increases the contrast of the image. Third, the p-wave is expanded by a beam expander (composed of lenses L2 and L3), and is transmitted to the light wedge (WL). The light on surface *B* of the WL interferes with the light reflected from the VM, so an interference pattern is generated. Fourth, the interference pattern arrives at the PBS. Since the light passes through the quarter-wave plate twice, the polarization direction of the p-wave is changed by 90 degrees. Since the detector plane of the camera is the image surface of surface B of WL, the interference pattern on surface B is formed on the detector plane of the camera. The distance between the detector plane of the camera and the focal plane of L4 is d, the calculation of which is shown in the Appendix A. Finally, the multi-field interference image is acquired using the camera. It should be noted that the QWP is placed near F2 so as to adjust the contrast of the interference image. The WL is the most critical element to generate the MFI, and its principle will be described in detail in the next section.

## 3. Principle of Multi-Field Interference

There are many situations where multiple beams’ interference is involved. Two classic examples are diffraction gratings [49] and plane-parallel plates [50]. In these two cases, only one interferogram is generated as the beams interfere. Our approach uses plane waves. In Figure 2, the red lines represent the plane waves from left to right, while the blue lines represent the plane waves from right to left. When light (*Li*) is incident on WL, refraction and reflection will take place one or more times on both surfaces of the WL. Then, the plane waves R1, R2 and R3 will be reflected, while T1, T2 and T3 will pass through the WL. Then, T2 is reflected by the VM and is again incident on the surface B of WL. Similarly, reflection will take place one or more times on both the surface B of the WL and the VM. Finally, R61, R62 and R63 will pass through the WL.

The two surfaces that interfere with each other are surface *B* of the WL and the VM. In the case of small angles, paraxial approximation is used in the following derivation of the relationship between reflection and refraction.

The reflection angle of R1 is equal to the incident angle of *Li* ($\theta_{1}$). The reflection angle of R2 ($\theta_{6}$) is given by
(1)$$\theta_{6}=2n\beta -\theta_{1},$$
where β and n represent the angle and the refractive index of WL, respectively. Similarly, the reflection angle of R3 ($\theta_{10}$) is given by:(2)$$\theta_{10}=4n\beta -\theta_{1},$$

$\theta_{4}$, $\theta_{8}$ and $\theta_{12}$ are the refraction angles of T1, T2 and T3, respectively, and are calculated as follows:(3)$$\theta_{4}=\theta_{1}-n\beta ,$$
(4)$$\theta_{8}=\theta_{1}-3n\beta ,$$
(5)$$\theta_{12}=\theta_{1}-5n\beta$$

Obviously, the angle between R1 and R2 and R2 and R3 is 2*nβ*, as is the angle between T1 and T2 and T2 and T3.

Let R3 be parallel to *Li*; then we have $\theta_{10}=\theta_{1}$. Then, the incidence angle of *Li*, $\theta_{1}$, is equal to 2nβ. According to Equation (3), we know that the refraction angle of T1 ($\theta_{4}$) will be equal to *nβ*, while according to Equation (4), we know that the refraction angle of T2 ($\theta_{8}$) will be equal to *−nβ*. If the VM is parallel to plane *B* of the WL, as shown in Figure 2a, the first order reflection angle of the rays of T2 reflected from the VM is *nβ*, and is parallel to T1. The T2 rays propagate through the WL and form the plane wave R61, so the angle of R61 ($\theta_{61}$) is equal to $\theta_{10}$. The second order reflection of T2 reflected from the VM is R62, and the third order reflection of the T2 reflected from the VM is the R63, where R61, R62 and R63 are all parallel to *Li* and R3, which will overlap when imaging on the camera. To obtain separate multi-field interference patterns, the VM needs to rotate counterclockwise by an angle of *α*. Then, according to the law of reflection, R61, R62 and R63 will rotate counterclockwise by angles of 2α, 4α and 6α, respectively, as shown in Figure 2b, and the opposite holds for clockwise rotation.

Finally, the reflection angles of R61, R62 and R63 are given by:(6)$$\theta_{61}=\theta_{10}+2\alpha ,$$
(7)$$\theta_{62}=\theta_{10}+4\alpha ,$$
(8)$$\theta_{63}=\theta_{10}+6\alpha$$

In other words, they will be separated by an angle 2*α* and interfere with R3 at the surface B of WL, where *z* = 0, so that multi MFI patterns are obtained. 

In order to limit the MFI in the range of one-half to three-quarters of the camera field of view, the following relationship must be satisfied, the calculation of which is shown in the Appendix A
(9)$$\frac{1}{12}\frac{f_{2}}{f_{3}f_{4}}ps\times re\leq \alpha \leq \frac{1}{8}\frac{f_{2}}{f_{3}f_{4}}ps\times re,$$
where $f_{2}$, $f_{3}$ and $f_{4}$ are the focal lengths of lenses L2, L3 and L4, respectively, *ps* is the pixel size of the camera and *re* is the resolution of the camera. 

The details of the interference are described below.

We define the amplitude reflection and transmission coefficients of WL’s surface *A* as $r_{A}$ and $r_{A}^{\prime }$, $t_{A}$ and $t_{A}^{\prime }$, the corresponding coefficients for surface B as $r_{B}$ and $r_{B}$, $t_{B}$ and $t_{B}^{\prime }$ and the amplitude reflection coefficient of the VM’s surface as $r_{VM}$. The primes indicate reflection or transmission from within the WL. 

So, the expression for the complex amplitude of R3 plane waves is
(10)$$E_{R3}(x,y,z,t)=t_{A}r_{B}^{\prime }r_{A}^{\prime }r_{B}^{\prime }t_{A}^{\prime }Ae^{i[\omega t+\Delta \phi_{R3}(x,y,z)]},$$
and the expression for the complex amplitude of R6j plane waves is:(11)$$\begin{array}{l} E_{6j}(x,y,z,t)\\ =t_{A}r_{B}^{\prime }r_{A}^{\prime }t_{B}^{\prime }t_{A}^{\prime }r_{VM}^{j}r_{B}^{j-1}t_{B}Ae^{i[\omega t+\Delta \phi_{R6j}(x,y,z)]} \end{array}$$

The calculations of Equation (10) and Equation (11) are shown in Appendix B.

The interference between R61 and R3, R61 and R62 and R62 and R63 is shown in Figure 3.

(1) In Figure 3a, beam R61 (p1-p2), which is the return of plane wave R61 from point p1 of the VM, interferes at point p2 with beam R3 (p2), which is the return of plane wave R3 from a position near point p2 on plane B of WL. The optical path difference of the two beams represents the local shape of the VM at point p1. The interference pattern formed by R61 and R3 is named S1.

(2) In Figure 3b, beam R62 (p1-p2-p3-p4), which is the reflection of plane wave R62 from point p1 on the VM, and through point p2 on the WL and point p3 on the VM, interferes at point p4 with beam R61 (p2-p3-p4), which is the reflection of plane wave R61 from point p2 of WL, and goes through point p3 on the VM. For these two beams, the optical path difference component caused by the local profile at point p3 of the VM is cancelled, and the remainder is the optical path difference corresponding to the local profile of p1. S2 is the interference pattern formed by R61 and R62.

(3) In Figure 3c, beam R63 (p1-p2-p3-p4-p5-p6), which is the reflection of plane wave R63 from point p1 on the VM and through point p2 on the WL, point p3 on the VM, point p4 on the WL and point p5 on the VM, interferes at point p6 with beam R62 (p2-p3-p4-p5-p6), which is the reflection of plane wave R62 from point p2 on the WL and through point p3 on VM, point p4 on the WL and point p5 on the VM. For these beams, the optical path differences caused by the local surface shapes at point p3 and p5 of the VM are cancelled, and the remainder is the optical path difference corresponding to the local shape of p1. As per previous cases, S3 is the interference pattern formed by R62 and R63.

Compared with S1, S2 will measure less the part between p3 and p1, while the distance between p3 and p1 is d1; compared with S2, S3 will measure less the part between p5 and p3, while the distance between p5 and p3 is d2. If the distance between VM and WL is *L*, then the distance between p3 and p1 is *L* (2*nβ* + 4*α*), and the distance between p5 and p3 is *L* (2*nβ* + 8*α*). According to the value range of *α*, its maximum is 0.23°. The refractive index of WL n is 1.5, and the wedge angle is 1°. When *L* is 10 mm, d1 = 0.68 mm and d2 = 0.84 mm. Since the short side length of VM is 40 mm, the influence of d1 and d2 on the whole mirror measurement is very small. As *L* decreases, d1 and d2 decrease accordingly, so the VM should be as close to WL as possible for the measurement.

The light intensity distribution of interference pattern S1 is calculated as follows:

The net complex amplitude at surface *B* of WL, where *z* = 0, is the sum of the complex amplitude of the R3 plane waves and the complex amplitude of R61 plane waves:(12)$$E_{sj}(x,y,t)=t_{A}r_{B}^{\prime }r_{A}^{\prime }t_{A}^{\prime }A\{r_{VM}^{j}r_{B}^{j-1}t_{B}e^{i[\omega t+\Delta \phi_{R61}(x,y)]}+r_{B}^{\prime }e^{i[\omega t+\Delta \phi_{R3}(x,y)]}\},$$

The resulting field intensity is
(13)$$I_{Sj}(x,y)=E_{Sj}(x,y,t)E_{Sj}^{*}(x,y,t),$$
where * denotes a complex conjugate.
(14)$$\begin{array}{l} I_{S1}(x,y)=\\ {\left(t_{A}r_{B}^{\prime }r_{A}^{\prime }t_{A}^{\prime }A\right)}^{2}\{{\left(r_{VM}^{1}\right)}^{2}{\left(r_{B}^{1-1}\right)}^{2}{\left(t_{B}t_{B}^{\prime }\right)}^{2}+{\left(r_{B}^{\prime }\right)}^{2}+2r_{VM}^{1}r_{B}^{1-1}r_{B}^{\prime }t_{B}t_{B}^{\prime }\cos[\Delta \phi_{R61}(x,y)\\ -\Delta \phi_{R3}(x,y)]\} \end{array}$$

Similarly, the intensity distributions of the interference patterns S2 and S3 are calculated as follows
(15)$$\begin{array}{l} I_{S2}(x,y)=\\ {\left(t_{A}r_{B}^{\prime }r_{A}^{\prime }t_{A}^{\prime }t_{B}t_{B}^{\prime }A\right)}^{2}\{{\left(r_{VM}^{2}\right)}^{2}{\left(r_{B}^{2-1}\right)}^{2}+{\left(r_{VM}^{1}\right)}^{2}{(r_{B}^{1-1})}^{2}+2r_{VM}^{2}r_{B}^{2-1}r_{VM}^{1}r_{B}^{1-1}\cos[\Delta \phi_{R62}(x,y)\\ -\Delta \phi_{R61}(x,y)]\} \end{array}$$
(16)$$\begin{array}{l} I_{S3}(x,y)=\\ {\left(t_{A}r_{B}^{\prime }r_{A}^{\prime }t_{A}^{\prime }t_{B}t_{B}^{\prime }A\right)}^{2}\{{\left(r_{VM}^{3}\right)}^{2}{\left(r_{B}^{3-1}\right)}^{2}+{\left(r_{VM}^{2}\right)}^{2}{\left(r_{B}^{2-1}\right)}^{2}+2r_{VM}^{3}r_{B}^{3-1}r_{VM}^{2}r_{B}^{2-1}\cos[\Delta \phi_{R63}(x,y)\\ -\Delta \phi_{R62}(x,y)]\} \end{array}$$
where the first and second terms are the intensities due to the two field *R3* and *R6j* individually, which form the background on the image. The interference effects are contained in the third term, where each $I_{Sj}$ produces an independent interference fringe Sj. 

We can draw two important conclusions from this result. First, the components $\Delta \phi_{R61}(x,y)-\Delta \phi_{R3}(x,y)$, $\Delta \phi_{R62}(x,y)-\Delta \phi_{R61}(x,y)$ and $\Delta \phi_{R63}(x,y)-\Delta \phi_{R62}(x,y)$ in the third term contain the information of the VM’s surface shape, which is what we are looking for. Second, we can adjust the stripe contrast ratio (SCR) through the parameters $r_{A}$ and $r_{A}^{\prime }$, $t_{A}$ and $t_{A}^{\prime }$, $r_{B}$ and $r_{B}^{\prime }$ and $t_{B}$ and $t_{B}^{\prime }$. The VM is the object to be measured. For simplification, and since the reflected beam is 180° out of phase as the beams with no primes are all reflected externally, we set $r_{VM}=-1$, and $r_{B}=-r_{B}^{\prime }$. The SCR can be expressed as: (17)$$SCR_{Sj}=\left\{ \begin{array}{l} \frac{-2t_{B}^{2}r_{B}^{\prime }}{{\left(t_{B}^{2}\right)}^{2}+{\left(r_{B}^{\prime }\right)}^{2}},j=1\\ \frac{2r_{B}^{\prime }}{{\left(r_{B}^{\prime }\right)}^{2}+1},j=2\\ \frac{2{\left(r_{B}^{\prime }\right)}^{3}}{{\left(r_{B}^{\prime }\right)}^{4}+{\left(r_{B}^{\prime }\right)}^{2}},j=3 \end{array}\right..$$

The minus sign indicates whether we are referring to a black or a white fringe. The amplitude of the fringe in the third term of expression 21 is:(18)$$A_{Sj}=\left\{ \begin{array}{l} 2r_{B}^{\prime }t_{B}{}^{2}{\left(t_{A}r_{B}^{\prime }r_{A}^{\prime }t_{A}^{\prime }A\right)}^{2},j=1\\ 2r_{B}^{\prime }{\left(t_{A}r_{B}^{\prime }r_{A}^{\prime }t_{A}^{\prime }t_{B}t_{B}^{\prime }A\right)}^{2},j=2\\ 2r_{B}^{\prime }{}^{3}{\left(t_{A}r_{B}^{\prime }r_{A}^{\prime }t_{A}^{\prime }t_{B}t_{B}^{\prime }A\right)}^{2},j=3 \end{array}\right..$$

The profile of $SCR_{Sj}$ and $A_{Sj}$ with the amplitude reflection coefficients $t_{B}$ are shown in Figure 4, where we have also assumed that there is no absorption in WL, so $t_{B}t_{B}^{\prime }+{\left(r_{B}\right)}^{2}=1$. We can draw an important conclusion from this figure: when $t_{B}$ ranges between 0.6 and 0.9, the interference effect is remarkable. The corresponding intensity transmission coefficients range from 0.36 to 0.81.

The constraint conditions to produce MFI using a laser pulse are summarized in two parts, as given below.

1. The third reflected wave vector R3 must be parallel to the incident wave vector Li, and the VM needs to rotate by an angle of *α*.

2. The corresponding intensity transmission coefficients of WL’s *B* plane must be within the range 0.36 to 0.81.

In the following sections, we present the experimental verification of our approach.

## 4. Results

### 4.1. Experimental Establishment

As shown in Figure 5, an experimental system was set up for multi-field interference imaging. In order to suppress the background light, all optical components were installed in a blackened box. The wedge of light was plated with an antireflection coating on side *A* with a transmittance of 0.98, while a portion of the transmission film was plated on side *B* with a transmittance of 0.73. The laser deployed is a model NPL52C from THORLABS.

The main parameters of the MFI imaging system are shown in Table 1.

The WL was mounted on a two-dimensional tilt adjustment mechanism (TDTAM), which allowed the easy adjustment of the pitch and azimuth. In order to make the reflected wave of R3 approximately parallel to the incident wave of *Li*, some necessary operations were performed. It should be noted that R1, R2 and R3—the waves in the propagation direction of the plane wave field—are difficult to recognize in the parallel optical path. Fortunately, they are easy to recognize in the convergent optical path, as light with different angles in the convergent optical path will converge on different locations on the focal planes. A paper screen was placed at the common focal plane of lenses L2 and L3 to observe R1, R2 and R3. Three corresponding light spots were produced in a straight line. When WL was rotated, the line connecting the three light spots was rotated accordingly, which represents the main section direction of the WL. A similar operation can be applied on the WL in order to move R3 closer to *Li* (see Figure 6b).

The VM was fixed on a one-dimensional turntable (ODT), with which horizontal adjustment of the VM was implemented. Meanwhile, pitch adjustment was achieved using some pads. Then, the *S* series light spots, including S1, S2 and S3, could be moved closer to R3 (see Figure 6c). When the ODT is fixed to the optical table, the phenomenon of MFI can be observed by fine-tuning the TDTAM and the viewing screen placed at the out-of-focus position of *f*_2_ (see Figure 6d).

### 4.2. Results and Discussion

The interference patterns obtained using the camera are shown in Figure 7. Three interference patterns, including S1, S2 and S3, were obtained in a single image. It should be noted that FG1 and FG2 in Figure 7a are two inherent patterns introduced by the surface reflection of the PBS. As shown in Figure 7b, when the vibration mirror is blocked, the interference patterns S1, S2 and S3 disappear, leaving only the inherent patterns. It can be seen from Figure 7a that S3 is not very clear, due to the low image contrast. However, the contrast of the interference patterns S1, S2 and S3 can be adjusted. As shown in Figure 7d, the contrast of S3 is obviously improved after enhancement.

Cross-sections of the interference patterns represented by three white lines in Figure 7c,d were extracted for further analysis. During the acquisition of the cross-sections of the interference patterns, the pulse frequencies of the laser were set to 200 Hz, 100 Hz, 50 Hz, 20 Hz and 10 Hz. As shown in Figure 8, the fluctuations of gray values in the curves are approximately consistent with the distribution in the interference patterns. The bottom of the curves corresponds to the black fringes in the interference pattern, while the peaks correspond to the white fringes.

Here, the contrast of the interference fringes is defined as:(19)$$SCR=\frac{SG_{\max}-SG_{\min}}{SG_{\max}+SG_{\min}-2offset},$$

Statistical results of the contrast are shown in Table 2. It can be concluded that the contrast of S1, S2, and S3 reaches more than 0.7 when the repetition frequency is 100 Hz or 50 Hz, but is relatively low at other repetition frequencies. Hence, the contrast of interference fringes is improved by adjusting the repetition frequency of the laser. When the frame rate of the camera is low (10 Hz) and the repetition frequency of laser is high (100 Hz or even higher), multiple pulses will be captured during one image frame. In this case, the brightness of the interference pattern will increase accordingly. In static cases, the interference pattern generated by the accumulation of multiple pulses does not cause crosstalk; however, crosstalk will occur under dynamic conditions. To avoid crosstalk, a single laser pulse is allowed in an image frame. In other words, the repetition frequency of a laser is equal to the frame rate of an image.

To verify the effectiveness of the MFI interference pattern, the same mirror was used for comparative testing. The interferometer used for comparison was a Fisba μshape2 HR phase-shift digital wave front interferometer, as shown in Figure 9c. As shown in Figure 9d, produced by the instrument’s software, the root mean square (RMS) value measured was 0.03 λ, which is 18.9 nm for λ=632.8 nm, while the RMS value measured using the method of this article was 20.3 nm. This is shown in Figure 9b, which was produced by the algorithm presented in Appendix C.

The difference between the two is very small, which proves that the MFI interference pattern can provide surface measurement information. The advantage of the proposed method is that it can produce three interference patterns simultaneously, which can be used for dynamic surface shape measurement while the VM is vibrating; the μshape2 HR interferometer does not have this ability [51].

## 5. Discussion and Conclusions

A multi-field interference imaging method is proposed to obtain the dynamic surface of the high-frequency vibrating mirrors. Compared with traditional interference methods, which produce only one interference pattern at a time, the proposed method can produce three interference patterns simultaneously at the surface *B* of the WL, which can be captured on a single image. A single laser pulse was applied in the MFI system and corresponding patterns were obtained on a single image. In this case, crosstalk was avoided perfectly, which is particularly desirable in dynamic applications. In summary, the MFI imaging method provides an effective way for dynamic surface measurement.

## Figures and Tables

**Figure 1 sensors-20-03372-f001:**
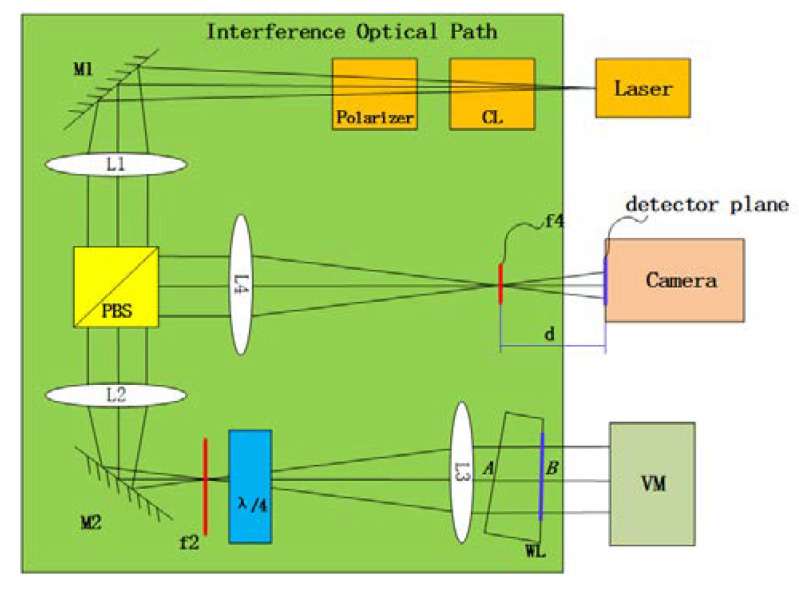
Schematic of multi-field interference (MFI) setup.

**Figure 2 sensors-20-03372-f002:**
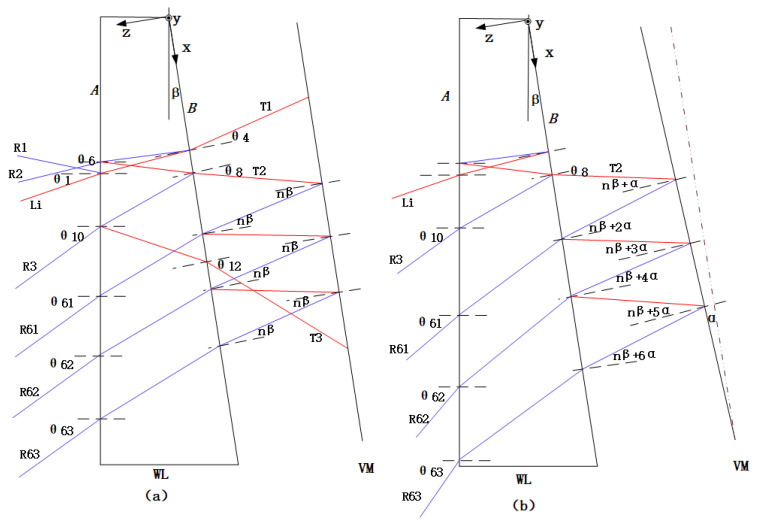
Light propagation between the light wedge (WL) and the vibrating mirrors (VM). (**a**) VM parallel to plane B of the WL; and (**b**) VM rotated counterclockwise by an angle of α.

**Figure 3 sensors-20-03372-f003:**
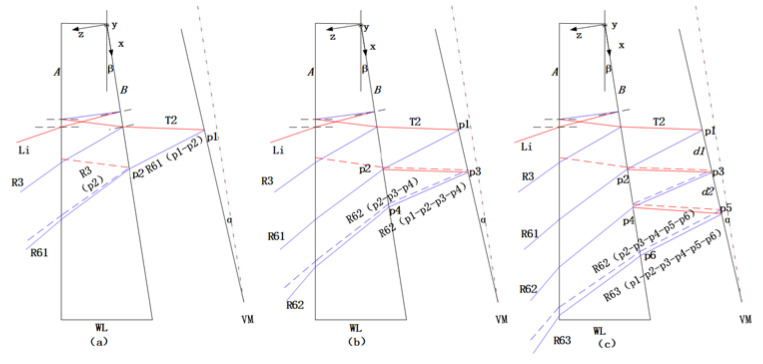
(**a**) Interference between R61 and R3; (**b**) interference between R61 and R62; and (**c**) interference between R62 and R63.

**Figure 4 sensors-20-03372-f004:**
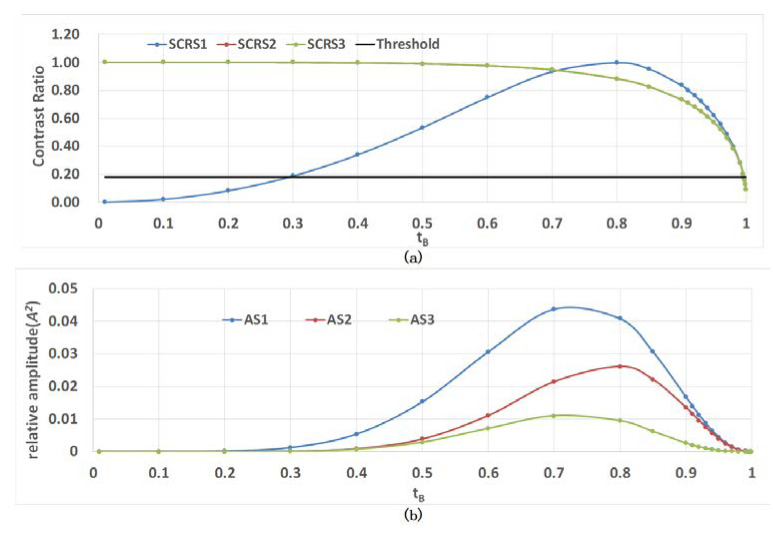
Relationship between ${\mathrm{SCR}}_{\mathrm{Sj}}$ and $A_{Sj}$ and the amplitude transmission coefficients $t_{B}$, SCR is the stripe contrast ratio and $A_{Sj}$ is the amplitude of the fringe. (**a**) Profile of ${\mathrm{SCR}}_{\mathrm{Sj}}$; and (**b**) profile of $A_{Sj}$.

**Figure 5 sensors-20-03372-f005:**
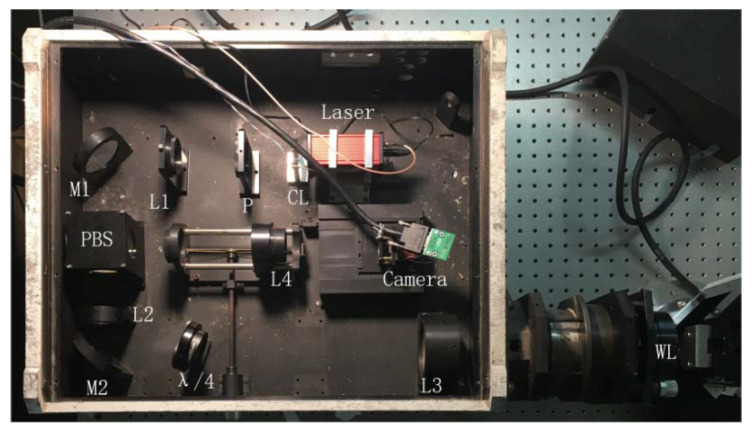
Layout of the optical path.

**Figure 6 sensors-20-03372-f006:**
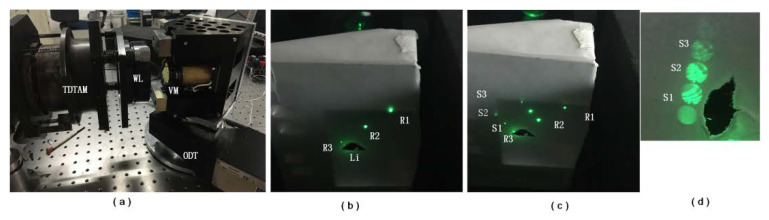
Adjustment process: (**a**) adjustment of WL and VM; (**b**) finding the focus point of the three reflected plane wave vectors R1 and R2 and R3; (**c**) finding the focus of the interference fields S1 and S2 and S3; and (**d**) MFI observed at the out-of-focus position.

**Figure 7 sensors-20-03372-f007:**
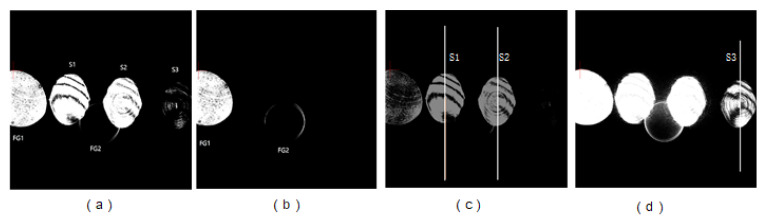
MFI patterns: (**a**) original image with resolution of 2048 × 2048; (**b**) Inherent patterns introduced by the PBS; (**c**) Contrast enhancement for S2; (**d**) Contrast enhancement for S3.

**Figure 8 sensors-20-03372-f008:**
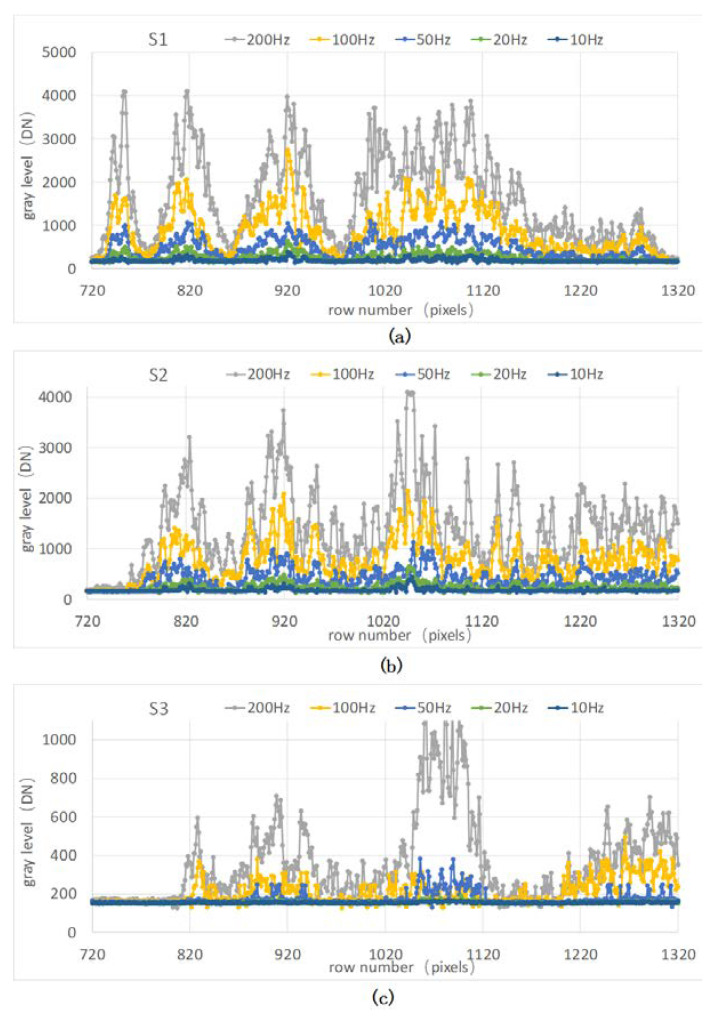
Cross-sections of the interference patterns in Figure 7c,d at different laser pulse frequencies: (**a**) cross-sections of S1; (**b**) cross-sections of S2; and (**c**) cross-sections of S3.

**Figure 9 sensors-20-03372-f009:**
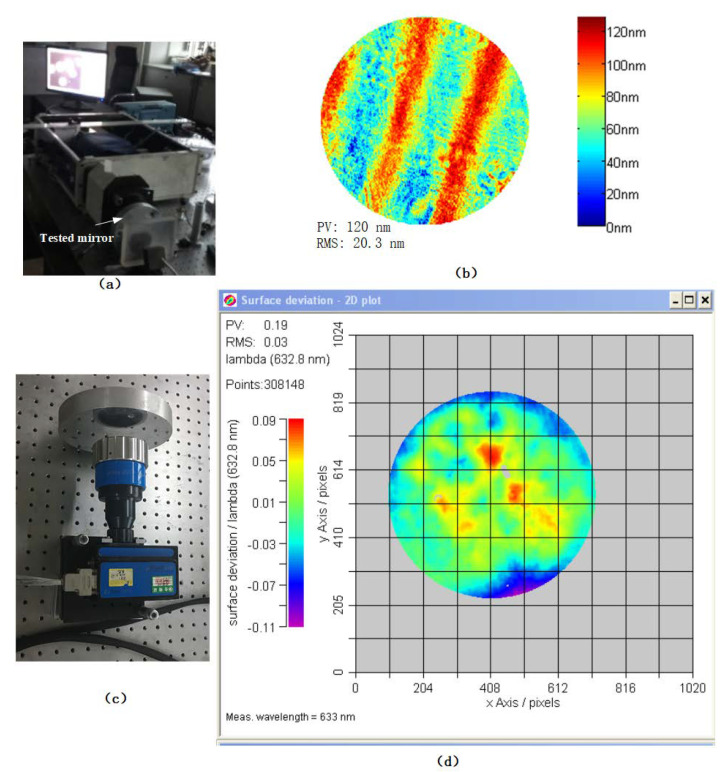
Comparative test. (**a**) MFI measurement; (**b**) result of the MFI; (**c**) μshape2 HR measurement; and (**d**) result of μshape2 HR.

**Table 1 sensors-20-03372-t001:** System parameters for MFI imaging.

Parameter	Value	Unit
Wavelength	532	nm
Repetition frequency	10–200	Hz
Pulse duration	6–129	ns
Pixel size	5.5	um
Resolution	2048 × 2048	pixels
Frame rate	10	fps
Integration Time	100	ms
Lens L1 focal length	200	mm
Lens L2 focal length	200	mm
Lens L3 focal length	380	mm
Lens L4 aperture	40	mm
Lens L4 focal length	180	mm

**Table 2 sensors-20-03372-t002:** Statistical results of the contrast. SG_max_: Max Gray (Unit: DN); SG_min_: Min Gray (Unit: DN); SCR: stripe contrast ratio; LRF: laser repetition frequency (Unit: Hz), DN: digital number (one grayscale unit).

LRF	S1			S2			S3		
	SG_max_	SG_min_	SCR	SG_max_	SG_min_	SCR	SG_max_	SG_min_	SCR
200	4095	331	0.91	3743	587	0.78	686	210	0.80
100	2054	300	0.85	2026	387	0.78	307	166	0.82
50	990	190	0.91	991	257	0.77	246	155	0.90
20	492	160	0.94	481	176	0.85	166	153	0.68
10	317	155	0.94	307	160	0.88	158	153	0.55

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
