# Peer review of "Multi-Field Interference Simultaneously Imaging on Single Image for Dynamic Surface Measurement"

_sensors, 2020, doi:10.3390/s20123372_

Round 1

Reviewer 1 Report

The paper is well organised and its readability is good. However, the reviewer has some comments as follows:

  1. Row 30 – There should be some references included which prove the statement about affecting imaging quality and modulation effect, for example:
  • Antonin Miks and Jiri Novak, „Analysis of imaging properties of a vibrating thin flat mirror,“ J. Opt. Soc. Am. A 21, 1724-1729 (2004)
  1. Row 50 – The reviewer suggests to use multiple-interferogram methods rather than three-interferogram methods only. See, for example:
  • Pramod Rastogi, Erwin Hack, Phase Estimation in Optical Interferometry, CRC Press, 2014.
  • Jiri Novak, Pavel Novak, and Antonin Miks, “Multi-step Phase-shifting Algorithms Insensitive to Linear Phase Shift Errors,“ Optics Communications 281 (21), 5302-5309 (2008)
  • Jiri Novak, “Five-Step Phase-Shifting Algorithms with Unknown Values of Phase Shift,“ Optik : International Journal for Light and Electron Optics 114 (2), 63-68 (2003)
  1. Rows 59-61 – References proving the statement about accuracy should be inserted.
  2. In the Introduction, there should be a more detailed discussion about the state-of-the-art of the measurement of vibrating mirrors and other optical components. All should be based on quantitative analysis and statements proven by citing appropriate references.
  3. Row 199 – There is a wrong reference on Figure 3, it should be Figure 4.
  4. Section 3 and 4 – Principle of interference patterns creation was well described. However, there is no information about the calculation of phase. More, there is no discussion about how the authors match images one to each other during calculation. This matching will affect the final accuracy. There is no analysis described, which should definitely be there. The final statement that the accuracy is similar to Fisba interferometer is not based on well-described quantitative data. This part of the paper has to be improved.

Based on the aforementioned notes the reviewer recommends the paper for publication after revisions.

Reviewer 2 Report

This manuscripts describes a multi-field interference imaging method to obtain the characterize the dynamic surface of high-frequency vibrating mirrors. By concurrently generating three interference patterns, the method combines the accuracy of three-interferogram approaches and the speed of single-interferogram methods. The work is suitable for publications in sensors if the author can address the following comments:

  1. The authors in general assumed too much knowledge from the audience. The introduction (lines 26-36) needs to provide a better explanation what the dynamics of surface is about. Does that mean a method is needed to tack the position/height of a flat vibrating mirror surfaces accurately, or is it for mapping the surface topology of a curved mirror?
  2. Lines 280-286 only stated that the method showed similar accuracy to the previous ushape HR. How is the proposed method better than previous work then? The advantage of the work needs to be stated explicitly during the comparison.
  3. The excess amount of equations significantly decreased the readability of the work. Consider moving some of them into the appendix, but only summarizing their biggest function in the main text.
  4. Some figures are poorly made. For example, figure 9 intends to compare the proposed MFI with traditional results, but the comparison was totally unclear because 9b and 9d presents the data in different scales and color maps. The text are also too small to see. Also, why the two RMS data were plotted in different formats (bar versus curves)?

Round 2

Reviewer 1 Report

The authors addressed all of the recommendations. Therefore, the reviewer recommends the paper for publication.